# Remote assessment in adults with Autism or ADHD: A service user satisfaction survey

**Marios Adamou**[1], **Sarah L. Jones** [2]*, **Tim Fullen**[2], **Nazmeen Galab**[2], **Karl Abbott**[2], **Salma Yasmeen**[2]

**1** University of Huddersfield, Huddersfield, United Kingdom, **2** South West Yorkshire Partnership NHS Foundation Trust, Wakefield, United Kingdom

* sarah.jones1@swyt.nhs.uk

**Data Availability Statement:** All relevant data are within the manuscript and its Supporting Information files.

**Funding:** The author(s) received no specific funding for this work.

## Abstract

Advances in digital health have enabled clinicians to move away from a reliance on face to face consultation methods towards making use of modern video and web-based conferencing technology. In the context of the COVID-19 pandemic, remote telecommunication methods have become much more common place in mental health settings. The current study sought to investigate whether remote telecommunication methods are preferable to face to face consultations for adults referred to an Autism and ADHD Service during the COVID-19 pandemic. Also, whether there are any differences in preferred consultation methods between adults who were referred for an assessment of Autism as opposed to ADHD. 117 service users who undertook assessment by the ADHD and Autism Service at South West Yorkshire NHS Partnership Foundation Trust from April to September 2020 completed an adapted version of the Telehealth Usability Questionnaire (TUQ). Results demonstrated that service users found remote telecommunication to be useful, effective, reliable and satisfactory. Despite this, almost half of service users stated a general preference for face to face consultations. There was no difference in the choice of methods of contact between Autism and ADHD pathways. Remote telecommunication methods were found to be an acceptable medium of contact for adults who undertook an assessment of Autism and ADHD at an NHS Service during the COVID-19 pandemic.

## Introduction

Advances in digital health have enabled clinicians to move away from a reliance on face to face consultation methods towards making use of modern video and web-based conferencing technology [1, 2]. Digital health is a widely used term that encompasses an enormous variety of products from consumer-focused mobile apps with no clinical validation to regulator-approved apps aimed at patients, physicians or clinical pathologists, to tools targeted at researchers. It also includes potentially disruptive technologies whose full impact has yet to be understood [3]. Proponents for the use of digital solutions in health care settings suggest that they have the potential to enhance patient choice, ensure cost efficiencies are maximised and provide a more flexible platform for healthcare delivery to patients [4].

**Competing interests:** The authors have declared that no competing interests exist.

In the field outside mental health, the use of remote and digital consultations has been and will continue to be, extensively researched as the clinical practices change and technologies evolve. A full review of the place of all remote and digital consultations in this area is outside the scope of this paper. Readers can access reviews according to the health conditions or medical practice according to their interest such as the ones for diabetes [5], surgical care [6] or chronic obstructive pulmonary disease [7]. A systematic review of the economic evaluations of telemedicine in various specialty areas found that telemedicine is cost-effective for applying in major medical fields such as cardiology but in dermatology, papers could not confirm the positive economic capability of telemedicine [8].

In the field of mental health there was once reticence to engage in remote telecommunication methods because psychiatrists, psychiatric nurses and clinical psychologists are more fundamentally interested in the human element of the patient interaction [9]. In contrast, patient interactions in physical health settings are necessarily more transactional in nature. Nonetheless, researchers have explored the potential benefits of this method of working in psychiatric populations. Cowpertwait and Clarke for example conducted a meta-analysis on the effectiveness of web-based psychological interventions for patients with depression [10] and found this form of intervention to be moderately effective in reducing depressive symptoms and improving well-being. Nevertheless, they also found significant heterogeneity in the results which was explained by the level of human engagement in each programme of intervention. Specifically, interventions which included face to face human engagement and feedback, produced higher effect sizes than ones which did not.

In terms of neurodevelopmental disorders, although they can be put under same diagnostic category conceptually [11], their core symptoms are different according to the diagnostic criteria. For example, Attention deficit hyperactivity disorder (ADHD) is characterised by severe deficits in attention, hyperactivity and impulsivity, whereas Autism spectrum disorder (ASD) is associated with impaired communication and social interaction skills, in addition to repetitive and restricted behaviour and interests (DSM-5) [12]. Although the above two disorders can co-exist [13] the needs of the patients throughout an assessment process are not the same. A rather recent systematic review evaluating the implementation of technologies to assess, monitor and treat neurodevelopmental disorders concluded that it is unclear whether there is sufficient evidence to support their use in clinical settings [14].

In terms of Autism, the literature does not have much to offer with regards to the role of digital health in the diagnosis of adults with Autism. In a systematic review Knutsen et al. [15] defined "telemedicine" as "the use of medical information exchanged from one site to another via electronic communications to improve a patient's clinical health status". This review brought together 35 papers with only two studies including an adult population. One study was a survey which included 45 adults (85% of which had an intellectual disability) and claimed "increased recognition of anxiety or mood disorders, symptom improvement, and more frequent adjustments in medication" [16]. The other study used a within-subjects crossover design and included 22 people. Although that study was not powered to show discriminative ability between the face to face and the remote version of Module 4 of the ADOS the study adopted, the authors claimed that "an autism assessment can be administered remotely with high levels of reliability using ADOS-2" [17]. A different scoping review in the use of telehealth for facilitating the diagnostic assessment of Autism identified 10 studies [18] and suggested that for certain presentations, remote assessment methods might be as reliable as face to face consultations when making a diagnosis [18]. This claim however cannot be extended at least to adults, since the only adult study the authors included was the Schutte et al. [19] study.

In terms of ADHD, the literature also does not have much to offer with regards to the role of digital health in the diagnosis of adults with ADHD. A recent literature review on the use of

telemedicine in the management of ADHD identified 11 studies [20], one of which included adults [21]. The adult study which sampled 129 adults referred to telemedicine visits conducted by specialists bringing significant improvements in participants' mental health status. All identified studies of the literature review used telemedicine as either augmentation to standard care, consultation (patient to provider or provider to provider), and evaluation; no study utilised telemedicine as an independent means for delivering direct clinical care to patients. The review identified extant literature on telemedicine in ADHD to be lacking.

In the context of the COVID-19 global pandemic, remote telecommunication methods have become much more common place in mental health settings. A study conducted at a University Clinic reported using the short form patient satisfaction questionnaire (PSQ-18) [22] that overall satisfaction with psychiatric care was high [23] whilst another observed that the frequency of patient contact within a community psychiatry service was maintained using remote telecommunication consultations and that prescribing practices were largely unaffected [24]. For adults, the COVID-19 related literature in Autism is not developed, although there is some discussion in the literature pertaining to children [25]. For adult ADHD, although there are initial reports of the effects of the pandemic and risk factors in adult populations [26, 27] there is again nothing specific to digital health.

The current study seeks to investigate (in a COVID-19 context) first whether remote telecommunication methods are preferable to face to face consultations for adults referred to an Autism and ADHD Service and second whether there are any differences in preferred consultation methods between adults who were referred for an assessment of Autism as opposed to ADHD.

## Methods

All patients who started an assessment by the ADHD and Autism Service at South West Yorkshire NHS Partnership Foundation Trust from April to September 2020 were eligible. This regional Neurodevelopmental Service provides diagnostic assessments for people over 18 years old who do not have a learning disability. Referrers select which pathway to refer their patients to using a form developed by the ADHD and Autism Service, which guides the collection of clinical information. Referrers can refer to both pathways simultaneously using different forms, however for the purposes of this study dual referrals were not considered. During this period, 49 assessments commenced for Autism and 113 for ADHD, but due to the COVID-19 restrictions, were conducted either using telephone or video conferencing. The video platform used was the "AccuRX" video consultation system (https://www.accurx.com). This platform securely sends a patient a link via SMS message which takes them to a secure video chat room with the clinician.

After their assessment, patients were invited by letter to complete a service user satisfaction survey in relation to their assessment experience using telephone or video. Enclosed with the letter was a paper version of the survey and a link to an online version. Participants were asked to return the paper copy via pre-paid envelope to the Trust Quality Improvement and Assurance Team (QIAT) or via accessing the online survey application by copying the link into an internet browser. This survey was approved by the SWYPFT Quality Improvement and Assurance Team as a Service Development initiative. This study was approved as a Service Improvement activity by the Quality Improvement and Assurance Team of South West Yorkshire Partnership NHS Foundation Trust. The reference number is: 20/21SE04. Participants provided informed verbal consent.

The survey questions were based on the Telehealth Usability Questionnaire (TUQ) [28] and are appended with this paper. The TUQ was designed to be a comprehensive

questionnaire which covers all usability factors, including *usefulness*, *ease of use*, *effectiveness*, *reliability*, *and satisfaction*. Practitioners at the ADHD and Autism Service adapted the TUQ questions for the purpose of this study. This was done using an iterative process of considering feedback and consensus reaching until the adapted version was agreed.

The survey takes approximately five minutes to complete and requires basic writing materials for the paper version or access to the internet to complete the online version. Respondents were asked to mark the appropriate box, corresponding to their experience of remote assessment with the Service.

The responses to the questions and selected demographic characteristics were collected by the Trust Quality Improvement and Assurance Team who imputed these in a Microsoft Excel spreadsheet. The spreadsheet was then presented for analysis. The demographic characteristics are listed in Table 1.

Respondents invited from the ADHD pathway were those who underwent an assessment using the Diagnostic Interview for ADHD in Adults (DIVA). The DIVA 2.0 was published by the DIVA Foundation in the Netherlands [29] and is a semi-structured interview based on the DSM-IV criteria for ADHD. The DIVA 2 evaluates all the DSM-IV criteria for adult ADHD based on the clinical judgment of the interviewer regarding the participant's answers and the accompanying person who knows the interviewee. The DIVA provides multiple examples for each criterion and can help for a better decision about the existence or non-existence of the symptoms. Finally, the interview assesses the functional impairment in five domains due to adult ADHD. This semi-structured interview takes approximately two hours to complete and was administered by clinicians (two medical doctors, two physician associates, two senior advanced nurse practitioners and one advanced nurse practitioner) who had received training in its administration.

Respondents invited from the Autism pathway were those who underwent an assessment for the purpose of collecting a full psychiatric history. The psychiatric history followed the scheme for history taking recommended by the Oxford Textbook of Psychiatry [30] and involves collecting information for the reason for referral, present condition, family history, personal history, present social situation, previous medical history, previous psychiatric illness, forensic history and personality before illness. Special emphasis was given in the developmental history component of the personal history. This assessment takes two to three hours and was conducted by specialist Autism Practitioners or a medical doctor.

For the analysis, descriptive and inferential statistics were explored using SPSS Version 26. Frequency (count and percentage) for each answer provided by respondents was calculated. Chi-squared goodness of fit test was employed to explore responses to each question with a significance level of 0.05 (5%) with assumed equal values. Further analysis was conducted to explore the influence of different service pathways, and of gender and age on responses received using Chi-square Test of Independence and Fishers-Freeman-Halton exact (where assumptions were violated).

## Results

### Study population

Of the 162 service users invited to complete the survey a total of 117 participants (72.2% response rate) returned it; 108 (92.3%) completed the paper version and 9 (7.7%) used the online link. Of the 117 participants, 20 (17.1%) were from the Autism pathway (40.8% response rate) and 93 (79.5%) from the ADHD pathway (82.4% response rate); there were four people who accessed both pathways and were not included in the response rate calculation per pathway. In terms of services accessed, 78 (66.7%) accessed diagnostic assessment, 34 (29.1%)

**Table 1. Demographic information of survey respondents.**

|  | Frequency | Percentage |
|---|---|---|
| **Dissemination** |  |  |
| Paper | 108 | 92.3 |
| Web based | 9 | 7.7 |
| **Respondents** |  |  |
| Service user | 98 | 83.8 |
| Carer/Unpaid carer/Voluntary/Community group | 4 | 3.4 |
| Other | 14 | 12 |
| **Sex** |  |  |
| Male | 78 | 66.7 |
| Female | 38 | 32.5 |
| Unspecified | 1 | 0.9 |
| **Ethnicity** |  |  |
| White/White British | 92 | 78.6 |
| Asian/Asian British | 3 | 2.6 |
| Black/African/Caribbean/Black British | 2 | 1.7 |
| Mixed/Multiple ethnic groups | 2 | 1.7 |
| Other ethnic groups | 7 | 6 |
| Not specified | 11 | 9.4 |
| **Age range** |  |  |
| 17–20 | 15 | 12.8 |
| 21–30 | 55 | 47 |
| 31–40 | 21 | 18 |
| 41–50 | 11 | 9.4 |
| 51–60 | 7 | 6 |
| 61–70 | 4 | 3.4 |
| Not specified | 4 | 3.4 |
| **Pathway** |  |  |
| ADHD | 93 | 79.5 |
| Autism | 20 | 17.1 |
| ADHD and Autism | 4 | 3.4 |
| **Service Accessed** |  |  |
| Diagnostic assessment | 78 | 66.7 |
| Medical review | 34 | 29.1 |
| Psychological intervention | 5 | 4.3 |
| **Remote method** |  |  |
| Telephone consultation | 87 | 74.4 |
| Video consultation | 25 | 21.4 |
| Telephone and Video consultation | 5 | 4.3 |

accessed a medical review, and five (4.3%) accessed psychological intervention. Of these, 87 (74.4%) used telephone consultation, 25 (21.4%) used video consultation, and five (4.3%) respondents had a combination of both telephone and video consultation. In terms of gender, 78 (66.7%) were male and 38 (34.5%) were female; one participant did not disclose their gender. In terms of ethnicity (taken from the 2011 UK Census categories), 92 (78.6%) respondents identified as White, three (2.6%) identified as Asian/Asian British, seven (6%) respondents identified as Other ethnic group, two (1.7%) identified as Black/African/Caribbean/Black, and two (1.7%) identified as Mixed/multiple ethnic, and 11 (9.4%) respondents did not specify

their ethnicity. In terms of age, 15 (12.8%) respondents reporting they were 17–20 years of age, 55 (47%) reported they were 21–30 years, 21 (18%) were 31–40 years, 11 (9.4%) were 41–50 years, seven (6%) were aged 51–60, four (3.4%) were aged between 61–70 years, and four (3.4%) chose not to disclose their age (see Table 1). There was no difference in the choice of methods of contact between pathways ($p$>0.05).

## Goodness of fit tests

In terms of *usefulness* of digital health methods, of the 117 participants recruited to the study, 81 (69.2%) were completely pleased to receive a remote appointment during the Covid-19 restrictions as an alternative to face to face consultations, 28 (23.9%) were pleased to some extent, 3 (2.6%) respondents were not sure, and 5 (4.3%) were not. A chi-square goodness-of-fit test was conducted to determine whether an equal number of responses were evident for each category. The minimum expected frequency was 29.3. The chi-square goodness-of-fit test indicated that responses were statistically significantly different from expected values ($\chi^2(3) = 135.274$, $p < .001$), with the majority of respondents pleased to receive a digital appointment. In terms of *ease of use and effectiveness*, a significant majority of respondents confirmed they felt they were able to communicate well during digital appointments ($\chi^2(3) = 43.718$, $p < .001$) (see Table 2), with a combined positive response of 76%. However interestingly, 61.5% ($n = 72$) (combined positive response) of the sample also suggested they felt they may have been able to better explain themselves if the consultation was face to face ($\chi^2(3) = 13.839$, $p = .003$). In terms of *reliability*, 69.2% ($n = 81$) of the sample felt that the clinician was able to complete a detailed assessment via remote contact methods ($\chi^2(2) = 72.974$, $p < .001$), and in terms of *satisfaction*, the same amount (69.23%) stated that they would tell others this was a good service ($\chi^2(3) = 123.991$, $p < .001$). However, also to be noted, 56 (47.9%) respondents stated a general preference for face to face consultation, compared to 33 (28.2%) who would not have preferred a face to face appointment ($\chi^2(2) = 11.436$, $p < .01$). Overall, 70 respondents (59.8%) thought that telephone and video conferencing appointments should be offered as an alternate option after Covid-19 restrictions have been lifted.

When asked to provide comments, positive feedback from respondents included that they were pleased at being offered appointments despite restrictions and liked the convenience of remote contact methods. In general respondents felt that clinicians were '*friendly*', '*understanding*' and '*polite*', providing ample opportunity for service users to explain themselves and to ensure a comprehensive initial assessment/review was completed despite the limitations of remote methods.

## Tests of independence

Fisher-Freeman-Halton exact test was used to explore association between service **pathways** (ADHD or Autism) and survey responses. There were no significant differences found between **pathways**, **type of remote assessment**, or **services accessed** ($p > 0.05$), *except* that those who accessed Psychological Interventions were more unsure as to whether they were pleased to receive a telephone/video appointment during the pandemic restrictions ($X^2(6) = 14.671$, $p < .05$). There were no significant difference found between **genders** ($p > 0.05$), *except* for responses pertaining to '*Do you think we should continue to offer telephone and video appointment after coronavirus restrictions are lifted*?' ($X^2(2) = 13.501$, $p < .001$), with 84.2% of females choosing '*Yes*' to the continuation of remote appointment methods post-pandemic, compared to only 48.72% of males.

**Table 2. Goodness of fit analysis.**

| Question | Response | Frequency | Percentage |
|---|---|---|---|
| Would you tell your friends and family that this is a good service?** | Yes | 81 | 69.2 |
| | Maybe | 18 | 15.4 |
| | No | 10 | 8.6 |
| | I don't know | 8 | 6.8 |
| Were you pleased to receive a telephone / video appointment during the coronavirus restrictions?** | Yes, completely | 81 | 69.2 |
| | Yes, to some extent | 28 | 23.9 |
| | No | 5 | 4.3 |
| | I don't know | 3 | 2.6 |
| Would you have preferred a face to face appointment?* | Yes | 56 | 47.9 |
| | No | 33 | 28.2 |
| | I don't know | 28 | 23.9 |
| Did you receive support during your appointment?** | Yes, from family | 55 | 47 |
| | Yes, from partner | 19 | 16.2 |
| | No | 43 | 36.8 |
| Do you think you would have been able to explain yourself better if you had been seen face to face?* | Yes, completely | 35 | 29.9 |
| | Yes, to some extent | 37 | 31.6 |
| | No | 33 | 28.2 |
| | I don't know | 12 | 10.3 |
| How well do you think you were able to communicate over the telephone / video call?** | Very well | 34 | 29.1 |
| | Well | 55 | 47 |
| | Not very well | 22 | 18.8 |
| | I don't know | 6 | 5.1 |
| Do you feel the clinicians completed a detailed initial assessment by telephone / video?** | Yes | 81 | 69.2 |
| | No | 8 | 6.8 |
| | I don't know | 28 | 23.9 |
| Do you think we should continue to offer telephone / video appointments after coronavirus restrictions are lifted?** | Yes | 70 | 59.8 |
| | No | 28 | 23.9 |
| | I don't know | 19 | 16.2 |

** $p < 0.001$,
* $p < 0.05$.

Significant differences were identified between **age** ($p < 0.05$) for three questions; '*Did you receive support during your appointment*?', with younger respondents receiving more support during remote appointment compared to the older respondents (see Fig 1). Also, differences according to age were found for '*How well do you think you were able to communicate over the telephone / video call*?' with 85.7% ($n = 18$) of total responses demonstrated the feeling that they were not able to communicate well, belonged to those aged 21–30 years (percentage within question). However, 65.5% ($n = 36$) of the same category suggested they were able to communicate well. Interestingly, a 100% of those aged 41–50 responded that they felt they were able to communicate well during the digital appointment (see Fig 2). Differences were also found when asked '*do you think we should continue to offer telephone/video appointments after coronavirus restrictions are lifted*? The majority of respondents in all age categories agreed that this should continue, however deviation from expected counts were most apparent for ages 21–30, with greater numbers ($n = 20$, 36.4%) of people suggesting digital appointments should not be offered after the restrictions lift (see Fig 3).

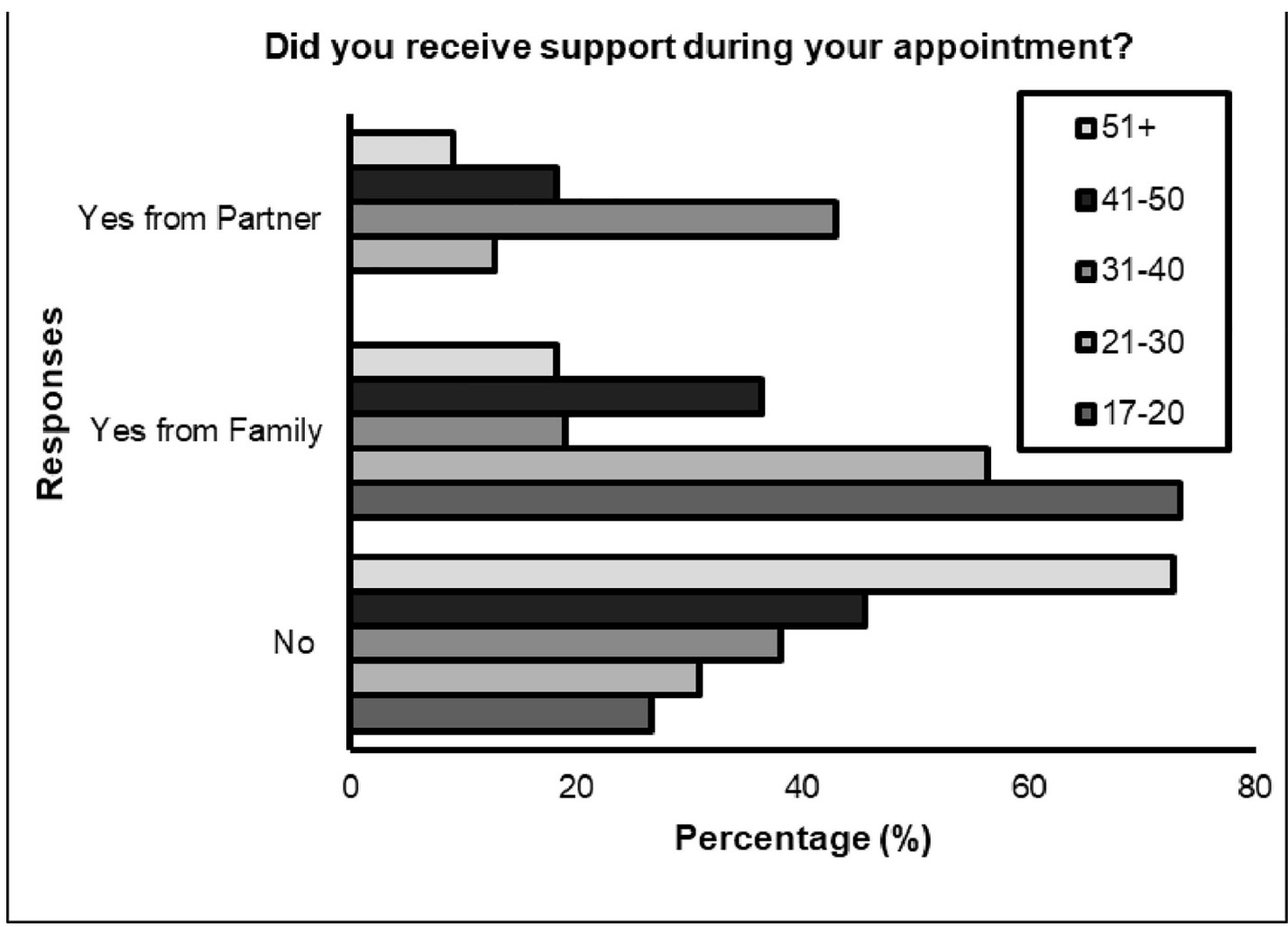

**Fig 1. Did you receive support during your appointment?**

## Discussion

The response rate of this study was higher than what we expected based on experience from previous Service surveys. On this occasion, the high response rate could be attributed to its judged salience to the respondents [31] who may have felt they wanted to support this initiative in the context of the pandemic. There has been little agreement on acceptable survey response rates among social scientists. Some accept rates as low as 30 percent; others reject anything below 70 percent [32]. With a response rate of higher than that our survey was valid. We have no information to explain the differential response rate between the pathways; one could have theorised that people undergoing an ADHD assessment would be less likely to post a survey due to their prevailing symptoms which includes disorganisation compared to Autism, but this is not what we found. Otherwise, the male predominance of our sample is what has already been reported for both Autism and ADHD [33, 34].

The way our Service delivered the remote assessments which were mostly by telephone, were reported to be useful, effective (with a caveat that more than half of the people said they felt they may have been able to better explain themselves if the consultation was face to face), reliable and satisfactory. Despite this, almost half of the people stated a general preference for

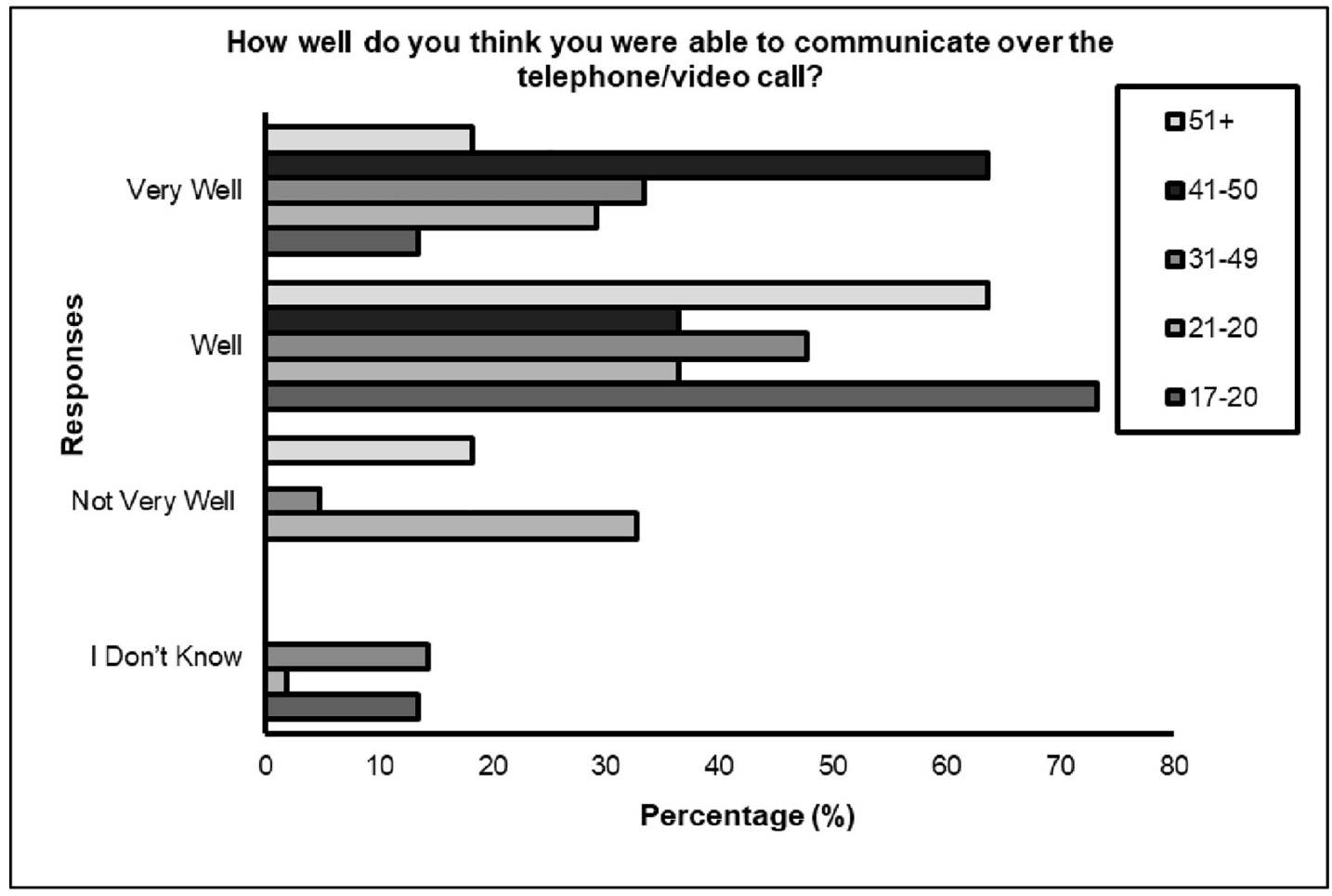

**Fig 2. How well do you think you were able to communicate over the telephone/video call?**

face to face consultations with the majority however suggesting that remote assessments could be reserved as an option for the future. Indeed, this is in line with a recent study that investigated patient and therapist experience with face to face (with face masks) compared to telepsychiatry sessions in a sample of adults with ADHD during the COVID-19 pandemic. Patients who took part in telepsychiatry sessions reported their experience to be 'less deep' than those who had face to face (with face masks) sessions [35].

We are very aware that the responses from our survey reflect the experience of people undergoing specific parts for a diagnostic assessment and not the complete assessment which would have included a diagnostic outcome. Future research should investigate if the diagnostic outcome (expected vs not expected by the patient) affects the way the remote assessment process is perceived. It is likely that the remote assessments will not be seen as useful, effective, reliable or satisfactory if the person feels that a face to face assessment would have generated a different diagnostic outcome.

Where we found difference to the responses was between genders, with females expressing the view that appointments should continue remotely after the pandemic restrictions. This preliminary finding supports the work of earlier research in non-clinical settings in which females were found to be more accepting and prolific in their use of mediated communication [36]. Given the relatively small number of female respondents in this study, caution must be

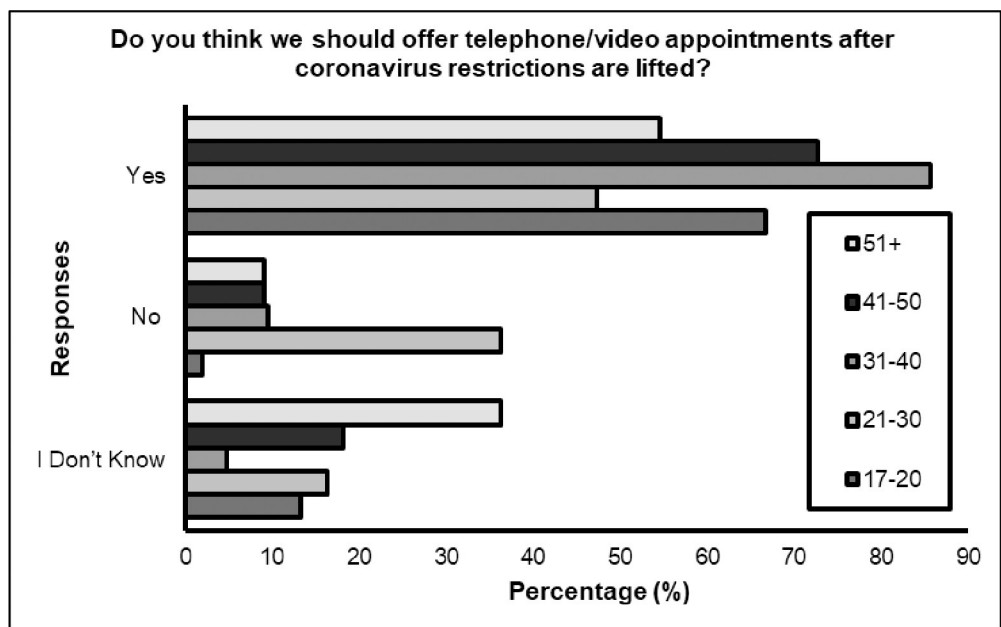

**Fig 3. Do you think we should offer telephone/video appointments after coronavirus restrictions are lifted?**

applied when interpreting this statistically significant result. Further research is required to determine whether this incidental finding represents a broader trend in different clinical sub populations.

We also found that younger people needed more support to proceed with the assessments as they found it more difficult to communicate well. This probably reflects the high level of need of people accessing this particular NHS Service, supported by younger patients feeling they required support from family or partners during assessments. This finding contrasts with the results of earlier research for patients from psychiatric outpatient settings which suggested that younger people are more accepting to health information technology [37].

In terms of suitability of remote assessments for the population group, we argue that at least for Autism, the diagnostic assessments should not be completed without a face to face (in person) evaluation. This is because the whole point of the assessment is to evaluate the person's ability to communicate. Taking into account the model of communication developed by Friedemann Schulz von Thun [38], such cannot be achieved during a remote evaluation. According to that model (which is known as the four-sides model), every message has four facets: fact, self-revealing, relationship, and appeal. Neglecting some of these sides increases the risk that sender and receiver of the message misunderstand each other particularly when sender and receiver come from different cultural backgrounds. This risk of such error during an Autism assessment is unacceptable. Remote assessments can indeed serve to collect the facts part of the message as on that level, the sender of the news gives data, facts and statements. However, in every message, there are another three parts to make communication complete. On the layer of the self-revealing or self-disclosure, the sender reveals himself/herself through conscious intended self-expression as well as unintended self-revealing; this cannot fully be achieved through even video solutions as one cannot get to know a person fully at a distance. Also, according to the four-sides model, the message has a relationship part. That part expresses how the sender gets along with the receiver and what he/she thinks about him. Depending on how he talks to him/her (way of formulation, body language, intonation) he/

she expresses esteem, respect, friendliness, disinterest, contempt, or something else. The ability to convey that during remote assessments but also construct it is also impaired. Finally, the appeal part of the message suggests that who states something, will also affect something. This appeal-message should make the receiver do something or leave something undone. The attempt to influence someone can be less or more open or hidden and this ability will be lost during remote assessments. Studies have already suggested that the use of remote methods in Autism might reduce diagnostic accuracy in more complex presentations because they do not give access to the full gamut of verbal and non-verbal cues which must be observed and interpreted [39]. For some cases specifically, for example in women with Autism where the phenomenon of 'camouflaging' has been suggested [40], making use of remote methods will make it even harder to conclude the diagnostic assessments remotely correctly as these women will be seen as unimpaired.

In terms of the suitability of assessments for adults with ADHD, it may indeed be possible to conduct remote assessments successfully as the symptoms alone are not the only requirement for diagnosis. In ADHD, there is symptomatic overlap with other disorders such as bipolar disorder [41] so emphasis is given to the psychiatric history which can be obtained remotely. Also, with the advent of artificial intelligence diagnostic solutions [42], the diagnostic process for ADHD can be even become more technologically reliant.

## Limitations

One of the limitations of the current study is that it is cross sectional in design which means that participant experience and opinions cannot be monitored over time. Also, the responses related to specific parts of a diagnostic process and not the complete diagnostic experience and were specifically linked to one NHS Service. A potential confound exists when considering the co-morbidity issues surrounding ASD and ADHD. Evidence suggests some overlap between ASD and ADHD, albeit not greatly [43]. Furthermore, the assessments specific to each pathway were not comparable, meaning that this could have a potential impact on effectiveness of remote assessment and patient preference. A final consideration is that the type of device (e.g., mobile phone, tablet, computer) or quality of the video and/or audio used by patients to access remote assessments was not surveyed. This could have a potential impact on service user experience. Future research into the acceptability of remote assessments can address these gaps by including a longitudinal design, compete assessment process (including diagnostic outcomes) and include more research sides.

## Conclusions

Remote telecommunication methods were found to be an acceptable medium of contact for adults who started an assessment of Autism and ADHD at an NHS Service during the COVID-19 pandemic. Both groups expressed a preference for face to face mode of assessment and particularly for Autism, that should be a clinical requirement. It may be that parts of the assessment can be conducted remotely particularly if it will increase access to Services.

## Supporting information

**S1 Appendix. The survey.**
(PDF)

**S1 Data.**
(XLSX)

## Author Contributions

**Conceptualization:** Marios Adamou.

**Formal analysis:** Sarah L. Jones, Nazmeen Galab.

**Methodology:** Karl Abbott.

**Resources:** Marios Adamou, Salma Yasmeen.

**Writing – original draft:** Marios Adamou, Sarah L. Jones, Tim Fullen.

**Writing – review & editing:** Marios Adamou, Sarah L. Jones.

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
