## [Decision Letter · Decision Letter 0]

12 Feb 2021

PONE-D-20-39665

Remote assessment in adults with Autism or ADHD: a service user satisfaction survey

PLOS ONE

Dear Dr. Jones

Thank you for submitting your manuscript to PLOS ONE. After careful consideration, we feel that it has merit but does not fully meet PLOS ONE’s publication criteria as it currently stands. Therefore, we invite you to submit a revised version of the manuscript that addresses the points raised during the review process.

We look forward to receiving your revised manuscript.

Kind regards,

Saeed Ahmed, MD

Academic Editor

PLOS ONE

2.Please provide additional details regarding participant consent. In the ethics statement in the Methods and online submission information, please ensure that you have specified (1) whether consent was informed and (2) what type you obtained (for instance, written or verbal, and if verbal, how it was documented and witnessed). If your study included minors, state whether you obtained consent from parents or guardians. If the need for consent was waived by the ethics committee, please include this information.

Review Comments to the Author

Reviewer #1: 1. Re: The 3 different types of assessments that were possible

a. Looking at patient preference for digital/ face to face appointment based on 'type of assessment' (diagnostic assessment/ review/psychological intervention) would be critical. The premise of the paper is that certain portions of the treatment process can be done virtually. This comparison seems essential to be able to draw a direct conclusions

b. It would be helpful to mention whether patients simply 'chose' a path (autism or adhd) or there was a triage process.

2. Was there any ADHD screening in the patients with Autism?

If the conclusion is that digital assessment maybe easier for 1 diagnosis than the other, this seems important

The co- morbidity of Autism and ADHD would make such distinction difficult to implement but possible to separate in a study environment.

3. One interview is semi- structured and the other is not. Is this correct?

If so, please comment on (or acknowledge the unknown factor) if this has a bearing on effectiveness of digital health

4. Majority of the assessments are via telephone. Is this the norm in local practice.

Having an A-V assessment compared to a telephone assessment would make a big difference in establishing the 4 aspects of the communication model described in the paper. If the n is not significant enough to comment on the 2 modalities separately, applying the findings to A/V assessments would be inaccurate.

5. Re: Conclusion

a. Both ADHD and Autism diagnosis are fraught with co-morbidities- hence a broad and thorough history taking is essential to both assessments- where the study states a potential role for digital health component.

b. high IQ and female gender compensates in both conditions hence some parts of the assessment should be face to face or A/v which maybe a close second.

6. The role of digital health maybe refinable based on the 'specific portion of assessment' (such as psychiatric history taking) but this data was not specifically looked at in the paper

If the effectiveness has to separate based on diagnosis- some form of triage process to ascertain the 'symptom cluster' of patients in each group, similar forms of assessment (structured vs non structured interviews) and some attempt at screening and removing overlap of diagnosis seem very important.

Currently,

Overall effectiveness seems difficult to ascertain since most people preferred face to face assessment.

Difference based on diagnoses does not seem to be supported by data

Difference based on type/ stage of assessment, if present but has not been pointed out in the paper.

Reviewer #3: Thank you for submitting your work to our journal. We would request the consideration and comments on the following:

1) Line 59 - 60, the authors state "...their interest such as the ones for diabetes (5), surgical care (6) or chronic

obstructive sleep apnoea (7)". Please double check the reference, is it for chronic obstructive sleep apnea or chronic obstructive pulmonary disease.

2) Line 60 - 63, the authors state "A systematic review of the economic evaluations of telemedicine in various specialty areas found that telemedicine is cost-effective for applying in major medical fields such as cardiology but in dermatology, papers could not confirm the positive capability of telemedicine (8)". Please consider elaborating on what the positive capability is. For example, is it referring to cost-effectiveness as for cardiology, or is it some other limitation of telemedicine other than economics?

3) Line 86 - 90, the authors state "A systematic review of “telemedicine” defined by the authors as “the use of medical information exchanged from one site to another via electronic communications to improve a patient’s clinical health status” brought together 35 papers (15) with only two studies including an adult population". Here the reader gets the impression that the study cited was conducted by the authors of this paper previous. Please consider rephrasing for the readers.

4) Line 110 - 112, the authors state "The review identified lacking in the extant literature on telemedicine in ADHD in the areas of assessment, diagnosis, or treatment of adults with ADHD". For ease if reading, consider rephrasing this sentence to: The review identified extant literature on telemedicine in ADHD to be lacking...

5) Line 120 - 122, please consider revisions in sentence structure.

6) Line 258, figure 2. The age grouping in figure 2 may need correction as they are stated as 17-20, 21-20, 31-19, 41-50, 51+. It may have been 21-30, 31-40, 41-50 etc.

7) Is there any particular reason for grouping ages as above, as opposed to using standard age groups adults and elderly?

8) Line 252 - 258, authors state "Also, differences according to age were found for ‘How well do you think you were able to communicate over the telephone / video call?‘ with 85.7% (n = 18) of total responses demonstrated the feeling that they were not able to communicate well, belonged to those aged 21-30 years (percentage within question). However, 65.5% (n = 36) of the same category suggested they were able to communicate well". Some clarification (perhaps in the discussion section) for this discrepancy between the same category stating first that they are not able to communicate well, then stating they were able to communicate well could prove to add to the texture of the manuscript.

9) Line 311 - 313, authors state "This finding contrasts with the results of earlier research for patients from psychiatric outpatient settings which suggested that younger people are more accepting to health information technology". Are there any hypothesis or plausible reasons for this observation?

10) In the discussion, it may be worthwhile to include the reasons for preference for remote assessments along with some discussion of the aspects of the providers experience as they to are the service user (provider), but this may be beyond the scope of the article.

Reviewer #5: In reference to this statement "We also found that younger people needed more support to proceed with the assessments as

310 they found it more difficult to communicate well. This probably reflects the high level of need

311 of people accessing this particular NHS Service", can you elaborate on what kind of support was needed to proceed with the assessments.

Did you have any exclusions about any subjects being on any psychotropic medications?

In regards to the limitations of the study:

Responses to the study received, did you look into any biases with respect to the influence of any family members' opinions when the subject was answering the questions.

Reviewer #6: The manuscript titled "Remote assessment in adults with Autism or ADHD: a service user satisfaction survey" is well designed, analyzed and executed.

The discussion and conclusions are well rounded and comprehensive.

The authors address the limitations well but the following points need to be addressed

- Was there any information collected relating to the type of platforms used by the participants in the study - example - Cellphones or Tablets vs Computers; Or Audio only devices (Telephone) vs Audio + video devices (handheld or computers)- Different age groups may be comfortable with certain types of devices used for accessing tele-psychiatric care which may impact their satisfaction. How is this accounted/adjusted for in the study?

- Were there any measures used to maintain uniformity of quality of the interaction between the service provider and the patient? Did the service provider and the patients have the same quality of Audio-Visual interaction/experience across devices/platforms - Some may have high quality video/audio while some may not. This would impact the user experience and therefore their rating of the experience. How is this accounted/adjusted for in the study?

- Stratification of the analysis by severity of ADHD/Autism may be important to assess if it plays a role in the experience of the assessment via Remote assessment. How is this accounted for in the study?

Few minor edits as follows:

- Line 102 - mention of reference (19) being "discussed above", but no discussion present pertinent to reference (19). Either a wrong reference or missing discussion

- Line 116 - "whilst" to "whilst"

- Line 121 - "reports the effects" is incomplete. It may be missing "of" or "on"

- Line 152/153 - The sentence is erroneous - either in structure/grammar. Not easy to discern the intended message of the statement.

---

## [Author Response · Author response to Decision Letter 0]

8 Mar 2021

Firstly, may we thank you the reviewers for the hard work in reviewing our manuscript. Your suggestions and comments have allowed us to reflect upon important considerations and these changes have served to strengthen our manuscript. Your contribution is greatly appreciated. 

Reviewer #1 

1. Re: The 3 different types of assessments that were possible

a. Looking at patient preference for digital/ face to face appointment based on 'type of assessment' (diagnostic assessment/ review/psychological intervention) would be critical. The premise of the paper is that certain portions of the treatment process can be done virtually. This comparison seems essential to be able to draw a direct conclusions. 

Response: Thank you for this comment. We agree that this comparison is important, within the analysis we found no differences in preference based on the type of assessment that patients experienced. We have added information about this in the results section. Please see tracked changes #1 (page 13, line 247). 

b. It would be helpful to mention whether patients simply 'chose' a path (autism or adhd) or there was a triage process. 

Response: Thank you for highlighting this. Patients do not have a choice of pathway; referrals are specific to pathways, we have added information about this in the methodology section, please see tracked changes #2 (page 6, lines 131-134). 

2. Was there any ADHD screening in the patients with Autism? 

If the conclusion is that digital assessment maybe easier for 1 diagnosis than the other, this seems important. The co-morbidity of Autism and ADHD would make such distinction difficult to implement but possible to separate in a study environment. 

Response: Thank you for your comment, this is an important consideration here. No, there was no ADHD screening within the ASD pathway. It is important to note that most patients within this study were under assessment (75%), therefore were yet to receive a diagnostic outcome. In terms of co-morbidity between ASD and ADHD, we don’t find that it is that high, but recognise that is a potential confound. We have therefore alluded to this in the discussion section, please see tracked changes #3 (page 17, lines 360-362). 

3. One interview is semi- structured and the other is not. Is this correct? If so, please comment on (or acknowledge the unknown factor) if this has a bearing on effectiveness of digital health 

Response: Thank you for highlighting this. Yes, this is correct. We have actioned this with a comment in the study limitations section. Please see tracked changes #4 (page 17, lines 363-364). 

4. Majority of the assessments are via telephone. Is this the norm in local practice. Having an A-V assessment compared to a telephone assessment would make a big difference in establishing the 4 aspects of the communication model described in the paper. If the n is not significant enough to comment on the 2 modalities separately, applying the findings to A/V assessments would be inaccurate. 

Response: Thank you for your comment, this is an important consideration. We explored preference based on the type of remote assessment, finding no difference in preference between those who received a telephone appointment and those who receive a video appointment, or a combination of the two. Within the context of our Service, face-to-face assessment is normal practice. We have added information regarding this in the results section, please see tracked changes #1 (page 13, line 247). 

5. Conclusion

a. Both ADHD and Autism diagnosis are fraught with co-morbidities- hence a broad and thorough history taking is essential to both assessments- where the study states a potential role for digital health component. 

Response: Thank you for the comment which we agree with.

b. high IQ and female gender compensates in both conditions hence some parts of the assessment should be face to face or A/v which maybe a close second. 

Response: Thank you for the comment which we agree with.

6. The role of digital health maybe refinable based on the 'specific portion of assessment' (such as psychiatric history taking) but this data was not specifically looked at in the paper. If the effectiveness has to separate based on diagnosis- some form of triage process to ascertain the 'symptom cluster' of patients in each group, similar forms of assessment (structured vs non structured interviews) and some attempt at screening and removing overlap of diagnosis seem very important. 

Response: Thank you for your comment. Triage process exists by referral source, as people are referred using a pathway specific referral form. We have added information to make clear to the reader, please see tracked changes #2 (page 6, lines 131-134). 

Currently, Overall effectiveness seems difficult to ascertain since most people preferred face to face assessment / Difference based on diagnoses does not seem to be supported by data / Difference based on type/ stage of assessment, if present but has not been pointed out in the paper.

Response: Thank you for your comments. We hope your concerns have been addressed in our revisions. 

Reviewer #3

1) Line 59 - 60, the authors state "...their interest such as the ones for diabetes (5), surgical care (6) or chronic obstructive sleep apnoea (7)". Please double check the reference, is it for chronic obstructive sleep apnea or chronic obstructive pulmonary disease. 

Response: Thank you for highlighting this. We have corrected this in text. Please see tracked changes #5 (page 3, lines 58-60). 

2) Line 60 - 63, the authors state "A systematic review of the economic evaluations of telemedicine in various specialty areas found that telemedicine is cost-effective for applying in major medical fields such as cardiology but in dermatology, papers could not confirm the positive capability of telemedicine (8)". 

Please consider elaborating on what the positive capability is. For example, is it referring to cost-effectiveness as for cardiology, or is it some other limitation of telemedicine other than economics?

Response: Thank you for highlighting this. This section discusses the economic capability. We have actioned this to make clear to the reader. Please see tracked changes #15 (page 3, line 63). 

3) Line 86 - 90, the authors state "A systematic review of “telemedicine” defined by the authors as “the use of medical information exchanged from one site to another via electronic communications to improve a patient’s clinical health status” brought together 35 papers (15) with only two studies including an adult population". Here the reader gets the impression that the study cited was conducted by the authors of this paper previous. Please consider rephrasing for the readers.

Response: Thank you for highlighting this. We have actioned this, rephasing this sentence to “In a systematic review Knutsen et al. (15) defined “telemedicine” as “the use of medical information exchanged from one site to another via electronic communications to improve a patient’s clinical health status”. This review brought together 35 papers with only two studies including an adult population.” Please see tracked changes #6 (page 4, lines 87-90).

4) Line 110 - 112, the authors state "The review identified lacking in the extant literature on telemedicine in ADHD in the areas of assessment, diagnosis, or treatment of adults with ADHD". For ease if reading, consider rephrasing this sentence to: The review identified extant literature on telemedicine in ADHD to be lacking... 

Response: Thank you for highlighting this. We have rephased this sentence as suggested. Please see tracked changes #7 (page 5, lines 109-110). 

5) Line 120 - 122, please consider revisions in sentence structure. 

Response: Thank you for highlighting this. We have changed sentence structure to read “For adults, the COVID-19 related literature in Autism is not developed, although there is some discussion in the literature pertaining to children (25). For adult ADHD, although there are initial reports the effects of the pandemic and risk factors in adult populations (26, 

27) there is again nothing specific to digital health”. Please see tracked changes #8 (page 5, line 117-120). 

6) Line 258, figure 2. The age grouping in figure 2 may need correction as they are stated as 17-20, 21-20, 31-19, 41-50, 51+. It may have been 21-30, 31-40, 41-50 etc. 

Response: Thank you for highlighting this. The figure is correct. Those in group over the age of 50 years were put together for aesthetic of the figure due to small group numbers. 

7) Is there any particular reason for grouping ages as above, as opposed to using standard age groups adults and elderly?

Response: Thank you for your comment. The reason for this was convenience for data collection. 

8) Line 252 - 258, authors state "Also, differences according to age were found for ‘How well do you think you were able to communicate over the telephone / video call?‘ with 85.7% (n = 18) of total responses demonstrated the feeling that they were not able to communicate well, belonged to those aged 21-30 years (percentage within question). However, 65.5% (n = 36) of the same category suggested they were able to communicate well". Some clarification (perhaps in the discussion section) for this discrepancy between the same category stating first that they are not able to communicate well, then stating they were able to communicate well could prove to add to the texture of the manuscript. 

Response: Thank you for this comment. Whilst this observation is interesting, we cannot explain why we observe this without further research.

9) Line 311 - 313, authors state "This finding contrasts with the results of earlier research for patients from psychiatric outpatient settings which suggested that younger people are more accepting to health information technology". Are there any hypothesis or plausible reasons for this observation? 

Response: Thank you for this question. We hypothesise this may be due to socialisation with technology.

10) In the discussion, it may be worthwhile to include the reasons for preference for remote assessments along with some discussion of the aspects of the providers experience as they to are the service user (provider), but this may be beyond the scope of the article.

Response: Thank you for highlighting this. Whilst we agree that this is an important area for discussion, we feel it is outside the scope of this paper. 

Reviewer #5

In reference to this statement "We also found that younger people needed more support to proceed with the assessments as they found it more difficult to communicate well. This probably reflects the high level of need of people accessing this particular NHS Service", can you elaborate on what kind of support was needed to proceed with the assessments. 

Response: Thank you for your comment. We have added a sentence here explaining that this observation was supported by the finding that younger patients required support from another person (family member or partner) during assessment. Please see tracked changes #9 (page 15, lines 315-317). 

Did you have any exclusions about any subjects being on any psychotropic medications? 

Response: Thank you for your comment. No, psychotropic medication was not an exclusion criterion. 

In regards to the limitations of the study: Responses to the study received, did you look into any biases with respect to the influence of any family members' opinions when the subject was answering the questions. 

Response: Thank you for this comment. Whilst an interesting consideration, this was not a part of the study design. 

Reviewer #6

The manuscript titled "Remote assessment in adults with Autism or ADHD: a service user satisfaction survey" is well designed, analyzed and executed. The discussion and conclusions are well rounded and comprehensive.

Response: Thank you for your generous comments. 

The authors address the limitations well but the following points need to be addressed

- Was there any information collected relating to the type of platforms used by the participants in the study - example - Cellphones or Tablets vs Computers; Or Audio only devices (Telephone) vs Audio + video devices (handheld or computers)- Different age groups may be comfortable with certain types of devices used for accessing tele-psychiatric care which may impact their satisfaction. How is this accounted/adjusted for in the study?

Response: Thank you for this comment. This was not surveyed; therefore, we are unable to comment. However, we recognise that this is a consideration and have added a sentence regarding this, in the limitations section of the manuscript. Please see tracked changes #10 (page 17, lines 364-367). 

- Were there any measures used to maintain uniformity of quality of the interaction between the service provider and the patient? Did the service provider and the patients have the same quality of Audio-Visual interaction/experience across devices/platforms - Some may have high quality video/audio while some may not. This would impact the user experience and therefore their rating of the experience. How is this accounted/adjusted for in the study?

Response: Thank you for this comment. Further to the response to the previous comment, clinicians and service users shared equal platforms but the quality of the video and/or audio was not surveyed or evaluated here, therefore we are unable to comment. However, we have now included this as a consideration in the limitations section. Please see as above. 

- Stratification of the analysis by severity of ADHD/Autism may be important to assess if it plays a role in the experience of the assessment via Remote assessment. How is this accounted for in the study?

Response: Thank you for your comment. Whilst an important consideration, within the context of this study, most of the patients included were under assessment for diagnosis. The diagnostic outcome was not finalised; therefore, we are unable to comment on severity of disorder.

Few minor edits as follows:

- Line 102 - mention of reference (19) being "discussed above", but no discussion present pertinent to reference (19). Either a wrong reference or missing discussion 

Response: Thank you for highlighting this, this has been actioned. Please see tracked changes #11 (page 5, lines 100-101).

- Line 116 - "whilst" to "whilst"

Response: Thank you for highlighting this, this has been actioned. Please see tracked changes #12 (page 5, line 114).

- Line 121 - "reports the effects" is incomplete. It may be missing "of" or "on"

Response: Thank you for highlighting this, this has been actioned. Please see tracked changes #13 (page 6, line 119). 

- Line 152/153 - The sentence is erroneous - either in structure/grammar. Not easy to discern the intended message of the statement. 

Response: Thank you for highlighting this. We have changed this statement to the following: “Practitioners at the ADHD and Autism Service adapted the TUQ questions for the purpose of this study. This was done using an iterative process of considering feedback and consensus reaching until the adapted version was agreed”. Please see tracked changes #14 (page 7, lines 153-155).

---

## [Editor Report · Decision Letter 1]

15 Mar 2021

Remote assessment in adults with Autism or ADHD: a service user satisfaction survey

PONE-D-20-39665R1

Dear Dr. Jones

We’re pleased to inform you that your manuscript has been judged scientifically suitable for publication and will be formally accepted for publication once it meets all outstanding technical requirements.

Kind regards,

Saeed Ahmed, MD

Academic Editor

PLOS ONE

---

## [Editor Report · Acceptance letter]

17 Mar 2021

PONE-D-20-39665R1 

Remote assessment in adults with Autism or ADHD: a service user satisfaction survey 

Dear Dr. Jones:

I'm pleased to inform you that your manuscript has been deemed suitable for publication in PLOS ONE. Congratulations! Your manuscript is now with our production department. 

Kind regards, 

on behalf of

Dr. Saeed Ahmed 

Academic Editor

PLOS ONE